



# Decadal changes in anthropogenic source contribution of PM$_{2.5}$ pollution and related health impacts in China, 1990–2015

Jun Liu[1], Yixuan Zheng[1], Guannan Geng[1], Chaopeng Hong[1], Meng Li[1], Xin Li[1], Fei Liu[2], Dan Tong[1], Ruili Wu[1], Bo Zheng[2], Kebin He[1,2], and Qiang Zhang[1,*]

[1] Ministry of Education Key Laboratory for Earth System Modeling, Department of Earth System Science, Tsinghua University, Beijing 100084, People's Republic of China
[2] State Key Joint Laboratory of Environment Simulation and Pollution Control, School of Environment, Tsinghua University, Beijing 100084, People's Republic of China

*Correspondence to*: Qiang Zhang (qiangzhang@tsinghua.edu.cn)

**Abstract.** Air quality in China has changed dramatically in response to rapid development of economy and policies. In this work, we investigate the changes of anthropogenic source contribution to ambient fine particulate matter (PM$_{2.5}$) air pollution and related health impacts in China during 1990-2015 and elucidate the drivers behind the decadal transition. We estimate the contribution of five anthropogenic emitting sectors to ambient PM$_{2.5}$ exposure and related premature mortality over China during 1990-2015 with 5-yr intervals, by using an integrated model framework of bottom-up emission inventory, chemical transport model, and the Global Exposure Mortality Model (GEMM). The national anthropogenic PM$_{2.5}$-related premature mortality estimated with GEMM for the nonaccidental deaths due to noncommunicable diseases and lower respiratory infections rose from 1.26 million (95% CI: 1.05, 1.46) in 1990 to 2.18 million (95% CI: 1.84, 2.50) in 2005; then, it decreased to 2.10 million (95% CI: 1.76, 2.42) in 2015. In 1990, the residential sector was the leading source of the PM$_{2.5}$-related premature mortality [559,000 (95% CI: 467,000, 645,900), 44% of total] in China, followed by industry (29%), power (13%), agriculture (9%) and transportation (5%). In 2015, the industrial sector became the largest contributor of PM$_{2.5}$-related premature mortality [734,000 (95% CI: 615,500, 844,900), 35% of total], followed by residential (25%), agriculture (23%), transportation (10%) and power (6%). The decadal changes in source contribution to PM$_{2.5}$-related premature mortality in China represents a combined impact of socioeconomic development and clean air policy. For example, active control measures have successfully reduced pollution from power sector, while contribution from industrial and transportation sector continuously increased due to more prominent growth of activity rates. Transition in fuel consumption dominated the decrease of contribution from residential sector. In the meanwhile, contribution from agriculture sector continuously increased due to persistent NH$_3$ emissions and enhanced formation of secondary inorganic aerosols under a NH$_3$ rich environment.

## 1 Introduction

Ambient air pollution is one of the most harmful environmental issues arising from development. It is a major risk factor to public health and is linked to various adverse health outcomes (Lim et al., 2012; GBD 2015 Risk Factors Collaborators, 2016).





Globally, ambient air pollution caused millions of deaths (Lelieveld et al., 2015; Cohen et al., 2017; Burnett et al., 2018), ranging from 4.2 (Cohen et al., 2017) to 8.9 (Burnett et al., 2018) million in 2015, depending on the adoption of risk functions. The associated economic costs were valued at between 3.8% (World Bank, 2007) and 9.9% of the GDP (World Bank, 2016). The largest number of deaths occurred in China (Cohen et al., 2017; Burnett et al., 2018), with a combination of severe air

pollution and high population density (Lelieveld et al., 2015; Liu et al., 2016c). Urgent actions are needed to reduce air pollution and improve public health (Zhang et al., 2012).

Driven by the rapid socioeconomic development and environmental policies (Zheng et al., 2018), air quality in China has changed dramatically over the past decades (Xing et al., 2015). Between 1990 and 2015 the country increased its total energy consumption by 3.4 times, thermal power generation by 7.6 times, pig iron production by 10 times, and civil vehicle population

by 28 times (National Bureau of Statistics, 2016). Consequently, China has experienced increasingly adverse impacts from worsening air quality (Xing et al., 2015) and associated diseases for decades (Lim et al., 2012; Cohen et al., 2017). Over large areas, the ambient concentrations of fine particulate matter with an aerodynamic diameter equal to or smaller than 2.5 micrometers ($PM_{2.5}$) far exceeded the World Health Organization (WHO) Air Quality Guidelines as well as the China's national air quality standards (Wang et al., 2014, 2015; Zhang and Cao, 2015).

In the meantime, China began responding to air pollution in the 1970s. The *Law on the Prevention and Control of Atmospheric Pollution* was formulated in 1987. Starting from 2005, national targets for sulfur dioxide ($SO_2$) and nitrogen oxides ($NO_x$) emission reductions were successively included in the *Eleventh* and *Twelfth Five-year Plans (FYP)*, and abatements have been achieved in the power, industrial, and transportation sectors in recent years (Huo et al., 2015; Liu et al., 2015, 2016a; van der A et al., 2017). In 2013, to tackle the severe and widespread air pollution, the Chinese Government launched the *National Air*

*Pollution Prevention and Control Action Plan*. Even more aggressive measures were implemented to reduce $PM_{2.5}$ concentrations by up to 25% in major metropolitan areas by 2017 (Zheng et al., 2018; Cheng et al., 2019).

The complex interactions between economic development and environmental regulations have caused air pollutant emissions over China to change significantly in the past twenty-five years. After growing rapidly during the early stages of the economic development (Lu et al., 2011; Zhao et al., 2013), $SO_2$ and $NO_x$ emissions (observed by satellite instruments) peaked in 2007

and 2011–2012 (Krotkov et al., 2016; Liu et al., 2016a; van der A et al., 2017). Studies on the historical $PM_{2.5}$ air pollution and related health impacts, and the decadal transition of source contributions provides opportunities to evaluate the effectiveness of past policies and point direction for prioritized control strategies in the future.

In China, annual estimates of mortalities attributable to $PM_{2.5}$ exposure ranged from 1.10 million to 1.37 million during 2010–2015 (Lelieveld et al., 2015; Liu et al., 2016c; Hu et al., 2017; Zheng et al., 2017), and studies have identified industrial and

residential sectors and coal burning activities as the leading contributors to premature deaths attributable to $PM_{2.5}$ during 2010-2013 (Hu et al., 2017; Ma et al., 2017; Gu et al., 2018). However, constrained by the availability of long-term emission datasets, limited efforts have been made to explore the decadal transitions of source contributions to $PM_{2.5}$ air pollution and related premature mortality in response to socioeconomic development and clean air policy. Recently, Zheng et al., (2019) has evaluated the emission source contributions to $PM_{2.5}$-related mortality during 2005-2015, and revealed the leading role of



agricultural and industrial sectors. But studies with longer temporal coverage to illustrate the decadal transition of different source sectors are still lacking. The changing patterns of sectoral emissions will result in varied source contributions at different stages of development (Zheng et al., 2018). Learning from the past is helpful for formulating future policies to optimize benefits. In this study, we combined the state-of-the-art Chinese long-term emission dataset with a regional air quality model and the

Global Exposure Mortality Model (GEMM) (Burnett et al., 2018), and investigate the decadal changes in anthropogenic source contribution of $PM_{2.5}$ air pollution and related health impacts in China from 1990–2015 for every five-year period. We quantify the contribution from the power, industrial, residential, transportation and agricultural sectors in each year, illustrate different transition routes for different source categories, and highlight the opportunities for further mitigation.

## 2 Materials and Methods

### 2.1 Bottom-up emission inventories

The bottom-up emissions inventory used in this study for mainland China during 1990–2015 was obtained from the Multi-resolution Emission Inventory for China (MEIC, http://www.meicmodel.org/). The emissions from other Asian countries and regions were taken from the MIX emission inventory (Li et al., 2017c). The MEIC model, developed by Tsinghua University, is a technology-based, bottom-up, anthropogenic emission model. By integrating a dynamic methodology with up-to-date

activity and local emission factors, it can provide model-ready emission inventories from 1990 to the present. The MEIC model includes more than 700 anthropogenic emission sources and quantifies emissions for ten pollutants and greenhouse gases: $SO_2$, $NO_x$, carbon monoxide (CO), nonmethane volatile organic compounds (NMVOC), ammonia ($NH_3$), carbon dioxide ($CO_2$), $PM_{2.5}$, $PM_{10}$, black carbon (BC), and organic carbon (OC). The on-road transportation emission was estimated at the county level and distributed across grids with the China Digital Road-network Map (Zheng et al., 2014). Emissions from coal-fired

power plants were constructed from the unit-based China coal-fired Power plant Emissions Database (CPED), which improved the spatial and temporal resolution of power emission (Liu et al., 2015). In addition, the model has an improved speciation framework to generate anthropogenic NMVOC emissions for various chemical mechanisms (Li et al., 2014). Model-ready emissions of $SO_2$, $PM_{2.5}$, $NO_x$, $NH_3$, CO, OC, BC, and NMVOC with Carbon Bond Mechanism version 5 (CB05) (Whitten et al., 2010) chemical mechanism were generated for the period from 1990 to 2015 in five-year intervals using MEIC and used

as the CMAQ input to simulate the $PM_{2.5}$ concentrations over China.

### 2.2 $PM_{2.5}$ exposure and source contributions

We used the Weather Research and Forecasting model (WRF) version 3.5.1 and the Models-3 Community Multiscale Air Quality model (CMAQ) version 5.0.1 (http://www.cmascenter.org/cmaq/) to simulate the $PM_{2.5}$ concentrations over China at a horizontal resolution of 36 km. Our WRF-CMAQ domain covers East Asia, including China, North and South Korea. Japan

and other Asian countries (Figure S1). The National Centers for Environmental Prediction Final Analysis data (NCEP-FNL) were used as meteorological initial and boundary conditions to drive the WRF model. Since NCEP-FNL data was only



available after 2000, for year 1990 and 1995, we used the meteorological data in 2000 to drive the WRF model. The meteorological fields simulated by the WRF and the emissions generated by the MEIC model were then used as the inputs to the CMAQ model. Biogenic emissions were calculated by MEGAN v2.1(Guenther et al., 2012). In the CMAQ model, the CB05 gas-phase chemistry and AERO6 aerosol chemistry with the ISORROPIA version II thermodynamic equilibrium module

(Fountoukis and Nenes, 2007) were used. The boundary conditions were generated with the global transport model GEOS-Chem (Bey Isabelle et al., 2001). For each year, the CMAQ model ran continuously for the entire year; the 7 days at the end of the previous December were used for model spin-up. Details concerning the model configuration were provided in our previous study (Zheng et al., 2015). To evaluate the performance of the WRF-CMAQ model, we conducted a detailed validation of the meteorological field simulations of the WRF and PM$_{2.5}$ mass concentrations by the CMAQ. More details are

provided in the Supporting Information (Table S1, Figure S2, and Figure S3).

In addition to CMAQ standard simulations between 1990 and 2015 at five-year intervals, we conducted five zero-out sensitivity simulations for each year by subtracting the emissions of each source sector from the total emissions. The source sectors included power, industry, residential, transportation, and agriculture. Then, the contribution of each source sector was determined by the difference between the standard and sensitivity simulations. Since we focused on the contribution and

relative importance of anthropogenic source sectors, we conducted another "clean" simulation for each year by removing all the Chinese anthropogenic emissions in the total model emissions, to exclude the contributions of boundary conditions, and emissions from biogenic sources, dust, sea salt, and other countries. Finally, to manage mass conservation, we implemented a grid-level normalization for the six sensitivity simulations to match the total PM$_{2.5}$ concentrations in the standard simulations.

**2.3 Estimates of health impacts attributable to long-term PM$_{2.5}$ exposure**

Long-term exposure to PM$_{2.5}$ has adverse impacts on human health; the most serious impact is death. In this study, we focus on the premature mortality due to the long-term PM$_{2.5}$ exposure. The premature mortality attributable to PM$_{2.5}$ can be determined by the application of the GEMM functions (Burnett et al., 2018). The GEMM was developed based only on cohort studies from 16 countries of outdoor air pollution that covers the global exposure range. It predicts hazard ratios between the long-term exposure to PM$_{2.5}$ and the nonaccidental deaths due to noncommunicable diseases and lower respiratory infections

(NCD+LRI), and deaths to five specific causes (5-COD). The hazard ratio function of the GEMM is described as Eq. (1).

$$GEMM(z) = exp\left\{\left[\theta ln\left(\frac{z}{\alpha} + 1\right)\right] / \left[1 + exp\left(-\frac{z-\mu}{\nu}\right)\right]\right\} \qquad (1)$$

Where $z = max\ (0,\ PM_{2.5} - cf)$, with $cf$ representing the counterfactual PM$_{2.5}$ concentration of 2.4 μg m$^{-3}$, below which no additional risk is assumed. ($\theta$, $\alpha$, $\mu$, $\nu$) are parameters describing the shape of the hazard ratio faction.

Following Burnett et al., (2018), we estimated the premature mortality attributable to PM$_{2.5}$ for both NCD+LRI and 5-COD.

The latter includes death due to ischemic heart disease (IHD), stroke, lung cancer (LC), chronic obstructive pulmonary disease (COPD), and LRI. The premature mortality attributable to PM$_{2.5}$ was calculated for population subgroups by year, age, sex,





and cause using Eq. (2), and the uncertainty and 95% confidence interval was estimated through the standard error (*SE*) of parameter $\theta$ in Eq. (1).

$$\Delta Mort_{yr,s,a,c}(z) = y_{0_{yr,s,a,c}} \times Pop_{yr,s,a} \times \left[1 - 1/GEMM_{a,c}(z)\right], \tag{2}$$

where $\Delta Mort_{i,j,k,l}(z)$ is the premature mortality attributable to $PM_{2.5}$ exposure level $z$ in year $yr$ for sex $s$, age group $a$, and

cause $c$; $y_{0_{yr,s,a,c}}$ is the baseline mortality rate in year $yr$ for sex $s$, age group $a$, and cause $c$; $Pop_{yr,s,a}$ is the population exposed to $PM_{2.5}$ in year $yr$ for sex $s$, and age group $a$; and $GEMM_{a,c}(z)$ is the hazard ratio associated with $PM_{2.5}$ exposure at level $z$ for cause $c$ *in* age group $a$.

The national baseline mortality rates of NCD, IHD, stroke, LC, COPD, LRI by age and sex, and population estimates by age and sex for each year were obtained from the Global Burden of Disease Project (Global Burden of Disease Collaborative

Network, 2017a, 2017b). The year-specific population distributions were obtained from the Gridded Population of the World version 4 (GPWv4) (http://sedac.ciesin.columbia.edu/data/collection/gpw-v4) (Doxsey-Whitfield et al., 2015), which has a horizontal resolution of 0.0083°×0.0083°. The 1990–2015 gridded population and annual average $PM_{2.5}$ concentration (36 km×36 km) simulated by CMAQ were then regridded to a uniform domain over China for mortality estimation at a horizontal resolution of 0.1°×0.1°. The $PM_{2.5}$-related premature mortality contributed by each sector was estimated using the gridded

relative source contributions determined by the CMAQ sensitivity simulations.

Besides, to identify the driving factors underlying the changes in long-term health impact, we calculated the contributions from four factors to the net changes in $PM_{2.5}$-related premature mortality between the neighboring years, i.e., population growth, population aging, baseline mortality rates and $PM_{2.5}$ exposure, to identify the driving factors underlying the changes in long-term health impact. Following the previous decomposition method (Cohen et al., 2017), we calculated the average factor

contributions through all the change sequences in the four factors.

## 3 China's air pollution regulations

In recognition of the importance of air pollution prevention, China has implemented air pollution controls since the 1970s. Table 1 lists the development sequence of the major air quality regulations in China, and Figure S4 shows the timetable of the emissions standards implemented in the major sectors during past decades. Historically, coal has been dominant in the energy

system, contributing between 60% and 70% of the primary energy. Coal burning has caused high levels of $SO_2$ and total suspended particulates (TSP) in the air. In 1987, the *Law on the Prevention and Control of Atmospheric Pollution* was formulated with the intent of reducing emissions from industry and from coal burning. Subsequently, the Law was revised in 1995 and again in 2000. During this period, the *Two Control Zones* (i.e., *Acid Rain Control* and *SO₂ Pollution Control Zones*) were established, and $SO_2$ emission controls were implemented to mitigate the acid rain and $SO_2$ pollution problems. A series

of emission standards was established in the power (GB13223-1991, GB13223-1996, and GB13223-2003) and industrial sectors (GB4915-1996, GB 9078-1996, GB4915-2004) to reduce the $SO_2$, particulate matter and $NO_x$ emissions (Figure S4).



In the transportation sector, China followed the vehicle emission standard system developed by the European Commission. Since 1998 China has taken a series of measures to address vehicle emissions, including implementing and updating vehicle emission standards for new vehicles, and phasing out old, high-emission vehicles (Huo et al., 2015). In 2000 and 2011, China implemented the national state I-emission standards for light-duty gasoline vehicles and heavy-duty diesel vehicles, respectively. Since 2005, national targets for reducing $SO_2$ and $NO_x$ emissions have been included in the Eleventh and Twelfth Five-year Plans (FYP). Installation of flue gas desulfurization (FGD) systems, selective catalytic reduction (SCR) and selective non-catalytic reduction (SNCR) equipment in coal-fired power plants and high-emission industries (such as iron and steel, cement), phase-out of small power plants, and more stringent vehicle emission standards have been mandated to achieve these targets (Liu et al., 2015, 2016a; van der A et al., 2017; Zheng et al., 2018). In 2012, $PM_{2.5}$ was included as an indicator in the new Ambient Air Quality Standards (GB 3095–2012). Subsequently, in 2013, China issued the *National Air Pollution Prevention and Control Action Plan* (hereafter the "*Action Plan*"), which for the first time made a commitment to reducing ambient $PM_{2.5}$ concentrations by up to 25% during 2013–2017. To fulfill the air quality targets, the government proposed ten pollution control measures and implemented a series of more stringent emission standards for the power, industrial, and transportation sectors (Figure S4) that have significantly reduced air pollutant emissions and substantially improved the air quality in China (Zheng et al., 2018; Cheng et al., 2019; Xue et al., 2019). The Thirteenth FYP was published recently. In addition to $SO_2$ and $NO_x$ emissions, control targets were also established for $NH_3$ and VOC emissions.

## 4 Results

### 4.1 Trends in annual average anthropogenic PM$_{2.5}$

As shown in Table 2, during 1990-2015, the $SO_2$, $NO_x$, $PM_{2.5}$, and $NH_3$ emissions in China increased from 13.6 Mt, 6.4 Mt, 8.9 Mt, and 7.2 Mt to 16.9 Mt, 23.7 Mt, 9.1Mt, and 10.5 Mt, with peaks occurring in 2005, 2010, 2005, and 2015, respectively. In response, the ambient $PM_{2.5}$ concentration over China has changed markedly during the 25-year study period. Figure 1 illustrates the annual average anthropogenic $PM_{2.5}$ concentrations in China from 1990–2015. The overall spatial pattern looked similar for all the years. High levels of $PM_{2.5}$ concentrations occurred in Northern, Central and Eastern China, covering most of the populous city clusters such as the Beijing-Tianjin-Hebei (BTH) region, the Yangtze River Delta (YRD), the Pearl River Delta (PRD), and the Sichuan-Chongqing region. Despite a short-term slowdown caused by the Asian Economic Crisis in 1997, a pronounced increase in nationwide $PM_{2.5}$ concentrations occurred during 1990–2005, driven by the dramatic growth of $SO_2$, $NO_x$, $NH_3$ and primary $PM_{2.5}$ emissions (Table 2). The population-weighted $PM_{2.5}$ concentration increased from 36.0 $\mu g\ m^{-3}$ in 1990 to 63.5 $\mu g\ m^{-3}$ in 2005. From 2005–2015, under the air quality regulations during the *Eleventh* and *Twelfth FYP* and the recent *Action Plan* (Table 1), the national $SO_2$ and $PM_{2.5}$ emissions were reduced by 49% and 33%, respectively (Table 2). During the same period, $NO_x$ emissions rose by 20%, but reached their peak during 2012–2013. As a result, the annual average $PM_{2.5}$ concentrations have decreased, and the population-weighted $PM_{2.5}$ concentration dropped to 49.9 $\mu g\ m^{-3}$ in 2015. During the entire period from 1990 to 2015, the effectiveness of control policies was offset by increased emissions from



expanding development; therefore, the annual PM$_{2.5}$ concentrations increased substantially in most of the regions. However, the more aggressive control measures resulted in improved air quality in some metropolitan areas, such as Beijing, the Yangtze River Delta, and the Pearl River Delta.

## 4.2 Trends in premature mortality attributable to PM$_{2.5}$

As shown in Figure 2, the national premature mortality attributable to anthropogenic PM$_{2.5}$ exposure estimated with GEMM NCR+LRI functions rose from 1.26 million (95% CI: 1.05, 1.46) in 1990 to 2.18 million (95% CI: 1.84, 2.50) in 2005 in 2005; then, it slightly decreased to 2.10 million (95% CI: 1.76, 2.42) in 2015. When adopting the GEMM 5-COD function, the excess deaths rose from 0.94 million (95% CI: 0.58, 1.24) in 1990 to 1.70 million (95% CI: 1.13, 2.13) in 2005; then, it decreased to 1.62 million (95% CI: 1.12, 2.01) in 2015. The mortality impacts of PM$_{2.5}$ exposure based on mortality rates of NCD+LRI were 27–33% higher than those based on mortality rates of 5-COD, indicating that the PM$_{2.5}$ exposure may contribute to mortality from causes other than the five-specific cause of death as we've known before (Burnett et al., 2018). The overall trend of PM$_{2.5}$-related premature mortality reflected the trend of national population-weighted PM$_{2.5}$ concentrations but exhibited a milder decline when the PM$_{2.5}$ concentrations fell after 2005, due to the impetus from other demographic factors (Figure 3). From 1990 to 2015, premature mortality caused by LRI remained relatively stable, and premature mortality caused by COPD declined, while premature mortality caused by IHD, LC and stroke increased, reflecting the demographic and epidemiological transitions over time (Yang et al., 2013). The rapid rise of these noncommunicable diseases poses challenges to China's public health.

The trends in premature mortality attributable to PM$_{2.5}$ are driven by changes in air quality, demographic factors, and baseline mortality rates. As shown in Figure 3, reductions in baseline mortality rates continuously contributed to the decrease of PM$_{2.5}$-related mortality, but these benefits were counterbalanced by increases resulting from population growth and aging. Before 2005, the deteriorating air quality contributed to the growth of PM$_{2.5}$-related mortality, whereas the notable improvements of air quality in past ten years have contributed to a decrease in PM$_{2.5}$-related mortality. Overall, however, the net reductions were rather small compared with the total premature mortality (for example, the reduction was only -2% from 2010–2015). With the population still growing and the accelerating aging of the population, it has become crucial to take further steps to sharply reduce PM$_{2.5}$ concentrations to effectively improve the related health benefits.

## 4.3 Source contributions to PM$_{2.5}$ air pollution and related premature mortality

We determined the source contributions from the power, industrial, transportation, residential and agricultural sectors to PM$_{2.5}$ air pollution and related premature mortality through a series of CMAQ sensitivity simulations. Figure 4 shows the spatial distribution of the PM$_{2.5}$ concentration contributed by each source sector in 1990, 2000, 2010 and 2015, and Figure 5 shows the relative and absolute source contributions to national population-weighted PM$_{2.5}$ concentration and related premature mortality from 1990–2015. The relative contributions of the source sectors to PM$_{2.5}$ concentrations and related premature mortality are similar (Table 2), and the nonaccidental mortality (NCD+LRI) was found to have enhanced statistical power to



characterize the shape of the GEMM functions compared to specific causes of death (Burnett et al., 2018). Therefore, we discuss the source contributions from a health perspective estimated with GEMM NCD+LRI functions in more detail, and the estimates with GEMM 5-COD were also listed in Table 2.

In general, the industrial sector is the prime source of PM$_{2.5}$-related premature mortality. This sector's contribution grew from 29% in 1990 to 37% in 2010, driven by the increasing demand for industrial production, but fell to 35% in 2015 under strengthened emission control measures during the *Action Plan*. The residential sector was the dominant source in 1990, but has had a decreasing trend over the past twenty-five years. The relative contribution of the power sector was almost stable from 1990 to 2005 but then began to decrease, owing to the active emission control over the past ten years. In contrast, the transportation and agricultural sectors have experienced increasing trends; their source contributions to PM$_{2.5}$-related mortality respectively increased from 5% and 9% in 1990 to 10% and 23% in 2015. These growing contributions indicate the necessity for attentions when planning future mitigations.

Historically, the power sector is the largest coal consumer, and is considered to be an important emission source in China. Since 1990, driven by the growing demand for electricity, the power sector has prominently increased its emissions, and became the leading source of SO$_2$ and NO$_x$ emissions in 2005 (Table 2). As a result, the population-weighted PM$_{2.5}$ concentration contributed by the power sector increased from 4.5 μg m$^{-3}$ in 1990 to 8.0 μg m$^{-3}$ in 2005. Sulfate, nitrate and ammonium (SNA) were the major chemical components, accounting for approximately 80% of the total PM$_{2.5}$ mass concentrations (Figure S5). Correspondingly, the absolute contribution of the power sector to PM$_{2.5}$-related mortality increased from 162,300 (95% CI: 135,400, 187,900) in 1990 to 287,000 (95% CI: 241,400, 329,500) in 2005. However, after 2005, a series of control measures were actively enforced in the power plants to meet the new emission standards (GB13223-2003, GB13223-2011), including installations of FGD systems in the 11[th] FYP, SCR and SNCR systems in the 12[th] FYP, and more stringent measures under the *Action Plan* (Zheng et al., 2018), which continuously reduced power plant SO$_2$ and NO$_x$ emission. Until 2015, the power sector accounted for only 130,900 (95% CI: 109,600, 150,900), or 6% of the PM$_{2.5}$-related deaths.

The industrial sector was the largest contributor to PM$_{2.5}$ air pollution and related mortality during 1995-2015, responsible for between 602,200 and 797,700 thousand (34–37%) deaths. Industrial emissions are emitted from both stationary combustion and industrial processes. Cement plants, iron and steel plants, and industrial boilers are the major contributors. Driven by the relentless growth of industrial productions, the industrial SO$_2$ and NO$_x$ emissions increased continuously from 1990 to 2010 (Table 2). But the industrial PM$_{2.5}$ emissions exhibit a different trend: they first increased from 1990-1995, then stabilized during 1995–2010, since the PM$_{2.5}$ emission mitigation from the switch of shaft kilns with precalciner kilns just counterbalanced the emission growth from other industrial sectors (Li et al., 2017b). Driven by those emissions changes, the population-weighted PM$_{2.5}$ concentration contributed by the industrial sector increased from 10.3 μg m$^{-3}$ to 22.1 μg m$^{-3}$ from 1990–2010 (Table 2), and the premature mortality contributed by the industrial sector increased prominently from 362,400 (95% CI: 302,600, 418,900) (29%) in 1990 to 797,700 (95% CI: 671,300, 915,300) (37%) in 2010. SNA and other unspeciated primary PM$_{2.5}$ emissions are the major chemical components of population-weighted PM$_{2.5}$, accounting for 42–58% and 31–48% of the mass concentration respectively (Figure S5). Until recently, the growing trend of industrial contribution has been





effectively reversed by a series of enhanced control measures under the *Action Plan*, including more stringent industrial emission standards, the elimination of outdated industrial capacity, the phasing out of small, polluted factories, and the elimination of small coal-fired industrial boilers (Zheng et al., 2018). In 2015, the industrial $NO_x$ emissions flattened and $SO_2$ and $PM_{2.5}$ emissions were remarkably reduced by 40% and 28% compared with their levels in 2010 (Table 2), which

consequently drove down the premature mortality shared by the industrial sector to 734,000 (95% CI: 615,500, 844,900) (35%). The residential sector is another major emitter of anthropogenic pollutants, including $PM_{2.5}$, BC, OC and NMVOC, due to poor combustion efficiency and lack of emission controls (Li et al., 2017b). The prime causes of residential emissions are consumption of fossil fuels and biofuels for heating and cooking. It is a major ambient air pollution source, especially during winter heating season in Northern China (Li et al., 2015; Liu et al., 2016b). On average, the residential sector was the second-

largest source of population-weighted $PM_{2.5}$ in China. The major chemical components are organic matter (OM) and BC, which compose approximately 46% and 16% of the residential contributed $PM_{2.5}$ concentration, respectively (Figure S5). In 1990, the residential sector was the leading contributor, accounting for 44% of the $PM_{2.5}$-related deaths, but it experienced an overall decreasing trend over time. There are several reasons for this decrease. First, China has undergone an accelerating urbanization process. Hundreds of millions of rural people have migrated from the countryside into the cities. The urbanization-

induced migration reduced emissions due to the switch from solid fuels to cleaner fuels after migration (Shen et al., 2017). Second, driven by the socioeconomic development, clean energy transitions from solid fuels to clean fuels such as natural gas and electricity gradually happened in rural households (Chen et al., 2016), resulting in a decrease in residential emissions. Third, from 2013–2017, China strove to replace the direct use of coal with electricity and gas-powered heating in millions of households in Northern China to mitigate air pollution in the countryside (Zheng et al., 2018; Cheng et al., 2019). Therefore,

the contribution of the residential sector gradually declined from 44% in 1990 to 25% in 2015 (Figure 5). The transportation sector is a growing source and is a major $NO_x$, NMVOC, and BC emitter. Before 2000, China was in the pre-stage of current vehicle emission standard system (Figure S4), and the emission control efficiencies were very limited in the transportation sector. Driven by vehicle population growth, $NO_x$ and $PM_{2.5}$ emissions increased by 167% and 229% from 1990–2000, and the population-weighted $PM_{2.5}$ contributed by the transportation sector increased from 1.7 $\mu g\ m^{-3}$ to 4.7 $\mu g\ m^{-3}$

$^3$ (Table 2). Nitrate and BC are the major chemical components, accounting for 55% and 16% of the $PM_{2.5}$ mass concentration contributed by the transportation on average (Figure S5). The $PM_{2.5}$-related mortality caused by transportation emissions increased from 60,300 (95% CI: 50,400, 69,700) (5%) in 1990 to 169,900 (95% CI: 142,700, 195,300) (9%) in 2000 (Figure 5). Then between 2000 and 2001, China implemented the national state I-emission standards for light-duty gasoline vehicles and heavy-duty diesel vehicles. Subsequently, the emissions standards were strengthened to stage II, stage III and stage IV

standards during 2003–2015 (Figure S4). With these policy measures, vehicle emissions did not increase as rapidly as the number of vehicles, and the VOC and $PM_{2.5}$ emissions from this sector have been declining since 2005 (Huo et al., 2015). $NO_x$ emissions have stabilized but are still growing. From 2000–2010, the contribution of the transportation sector to $PM_{2.5}$-related mortality was relatively stable; however, by 2015, when pronounced reductions were achieved in the power and industrial





sectors, the prominence of transportation sector increased, with a contribution of 218,300 (95% CI: 183,100, 251,200) PM$_{2.5}$-related deaths, or 10% of the total, revealing a growing threat to public health.

The agricultural sector is another growing source with surge of NH$_3$ emissions. During 1990-2015, the PM$_{2.5}$-related mortality shared by agricultural sector increased from 117,800 (95% CI: 98,500, 136,000) (9%) to 484,400 (95% CI: 406,500, 557,400)
(23%) (Figure 5). This increase was not only driven by the rising NH$_3$ emissions but also by the abrupt growing NO$_x$ emissions. NH$_3$ is an important precursor of ambient PM$_{2.5}$ that exists in the secondary forms of ammonium sulfate and ammonium nitrate. Over 90% of the NH$_3$ emissions stem from agricultural activities, including synthetic nitrogen fertilizer applications and livestock manure management. Because of the increasing demand for agricultural products and the lack of effective control measures, agricultural NH$_3$ emissions increased from 6.7 million tons in 1990 to 9.7 million tons in 2015. Consequently, the
population-weighted PM$_{2.5}$ concentration contributed by the agricultural sector increased from 3.4 µg m$^{-3}$ to 12.6 µg m$^{-3}$ during 1990–2010. However, it subsequently decreased to 11.6 µg m$^{-3}$ in 2015 due to NO$_x$ emission reductions imposed by the *Action Plan* (Table 2). Ammonium, nitrate and sulfate were the major chemical components of the agricultural population-weighted PM$_{2.5}$, accounting for over 98% of the mass concentration (Figure S5). Sulfate and nitrate are formed through oxidation and neutralization of precursor gases SO$_2$, NO$_x$, and NH$_3$. (NH$_4$)$_2$SO$_4$ is the preferential species due to its stability; the semivolatile
NH$_4$NO$_3$ is formed when excess NH$_3$ is available beyond sulfate requirements (Seinfeld et al., 2006). In the sensitivity simulations, when the agricultural NH$_3$ emissions were removed, the most nitrate formation was inhibited due to lack of NH$_3$. In contrast, sulfate can exist in the form of acid particles. Therefore, the amount of nitrate is much higher than the amount of sulfate in the agriculture-contributed PM$_{2.5}$ (Figure S5).

## 5 Discussion

### 5.1 Comparison with other studies

We compared our estimates of anthropogenic source contributions to national PM$_{2.5}$-related premature mortality with those of previous studies in China (Table 3). Most of the studies focused on one particular year for the period of 2010-2014 (Lelieveld et al., 2015; Hu et al., 2017; Gu et al., 2018; Reddington et al., 2019), and one study investigated the changes of source contributions during 2005-2015 (Zheng et al., 2019), whereas our study has a longer temporal coverage of 25 years, from 1990
to 2015. The differences between studies could arise from the differences in emission inventories and air quality models. The residential sector was highlighted in all the studies, sharing a contribution ranging from 15% (Zheng et al., 2019) to 42% (Reddington et al., 2019). The industrial sector was also identified as the leading source in five out of the six studies. A decreasing trend of the contribution from the power sector during 2005–2015 was illustrated by all the studies. Large discrepancies occurred in the agricultural sector, with the contribution ranging from 0.1% (Reddington et al., 2019) to over
30% (Lelieveld et al., 2015; Zheng et al., 2019). The magnitude of the contribution from the transportation sector was relatively close in all the studies. Overall, our estimates of relative source contributions were generally consistent with previous studies and well within the ranges of their estimates, and were closest to those from Hu et al., (2017), who estimated the same order





of source contributions with our study, i.e. industry, residential, agriculture, power and transportation, from the highest to the lowest.

## 5.2 Uncertainties and limitations

Our study has a number of uncertainties and limitations. First, we estimated the long-term exposure of PM$_{2.5}$ using CMAQ
model simulations rather than satellite-based estimates. In practice, the satellite-based estimates utilize information from satellite and observations, resulting in improvements in estimates (Ma et al., 2016; van Donkelaar et al., 2016; Xue et al., 2017). However, satellite remote sensing data was unavailable prior to 1998. If we had combined the CMAQ simulations for 1990–1995 and satellite-based estimates for 2000–2015, we would have introduced additional uncertainties, which may have perturbed the long-term trends of source contributions to PM$_{2.5}$, and weakened the information conveyed from the decadal
sectoral emission changes. To illustrate the impacts, we calculated the PM$_{2.5}$-related mortality with both CMAQ simulations and satellite-based estimates of van Donkelaar et al., (2016) and Ma et al., (2016) for three overlapping years: 2005, 2010 and 2015. The results showed that the PM$_{2.5}$-related premature mortalities estimated with the CMAQ simulations were comparable to those using satellite-based PM$_{2.5}$ (Figure 6). These differences were acceptable considering the uncertainty ranges from the GEMM functions. Previous studies on long-term trends in the chemical composition of population-weighted PM$_{2.5}$ (Li et al.,
2017a), the attribution of source contributions to the PM$_{2.5}$-related  mortality to emissions changes (Lelieveld et al., 2015; Ma et al., 2017; Gu et al., 2018; Zheng et al., 2019) also applied the chemical transport model simulations to estimate exposure.

Second, the uncertainty of this study also stems from the PM$_{2.5}$ exposure-response functions. Globally, the excess mortality associated with PM$_{2.5}$ air pollution in 2015 ranged from 4.2 million (Cohen et al., 2017) to 8.9 million Burnett et al., (2018), depending the choice of exposure-response functions. Burnett et al., (2018) constructed the up-to-date GEMM functions based
only on cohort studies of outdoor air pollution which covers the global exposure range, and estimated 2.47 million (GEMM NCD+LRI) and 1.95 million (GEMM 5-COD) deaths in China due to ambient PM$_{2.5}$ exposure. Our calculation with the GEMM risk functions illustrated 2.10 million and 1.62 million excess deaths related to the anthropogenic PM$_{2.5}$ exposure in 2015 by GEMM NCD+LRI and GEMM 5-COD functions. Our results were rather close to the estimates of Burnett et al., (2018), but prominently higher than the previous estimates of 0.87–1.37 million during 2010–2015 (Lelieveld et al., 2015; Liu et al., 2016c;
Hu et al., 2017; Zheng et al., 2017, 2019), which applied the Integrated Exposure-Response (IER) function developed in the Global Burden of Disease (GBD) Project (Burnett et al., 2014; Cohen et al., 2017). We conducted a sensitivity analysis by repeating the calculations with the IER risk functions used in GBD 2015, and estimated the anthropogenic PM$_{2.5}$-related mortalities in China in 2015 were 0.97 million (Table 4), which were close to the estimate of 0.87 million in 2015 by Zheng et al., (2019). The sensitivity analysis shows that different PM$_{2.5}$ exposure-response functions exhibit large discrepancies in
the estimates of PM$_{2.5}$-related mortality, which calls for more additional cohort studies in the China to improve the health risk models and narrow the gaps between different studies. However, since our study focus more on the relative fraction of source contributions to the population-weighted PM$_{2.5}$ and PM$_{2.5}$-related mortality, the results and conclusions won't change no matter which version of exposure-response functions we choose.



Third, to present the long-term trends in PM2.5-related premature mortality, we utilized the year-specific national population age structure and the cause-specific mortality shared in GBD 2016 (Global Burden of Disease Collaborative Network, 2017a, 2017b). However, there are substantial provincial heterogeneities in age structures and mortality due to differences in social economics, geographic distributions, and health services that were not studied in this work. The existing data from statistics

and other sources are not sufficient to provide such details.

Fourth, we differentiated the contributions from various source categories with a series of zero-out sensitivity simulations. To address the mass nonconservation resulting from the zero-out method, we normalized the sensitivity simulations to match the total PM2.5 concentrations in the standard simulation, which could introduce uncertainties. In fact, the zero-out method estimates the upper value for the contribution from gaseous precursors to secondary PM2.5, especially for the agricultural sector,

where NH3 dominates most of the emissions. The agricultural contribution estimated in this study includes the nonagricultural nitrate. According to our estimation, the agricultural sector accounted for 21–23% of the total premature mortality during 2010–2015, which is higher than the estimation (15%) using the source-oriented CMAQ model (Hu et al., 2017), but lower than the estimation (32-25%) that used the zero-out method in the global chemical transport model (Lelieveld et al., 2015) and the CMAQ model (Zheng et al., 2019) (Table 3).

**5.3 Policy implications**

Ambient PM2.5 pollution contributed substantially to premature mortality in China. To mitigate air pollution problems, the government launched air quality regulations decades ago. The transition of source contributions calls for adjustments to policy focuses on emission sectors.

Pronounced emissions abatements have been achieved in the power sector. Recently, the penetration rates of FGD and

SCR/SNCR systems has reached over 95%, and in 2017 approximately 71% of current power plants operated close to the level of "ultralow emission" (Zheng et al., 2018). Thus, the potential to reduce future emissions is limited in the power sector. Currently, industry is the largest contributor. To fulfill the air quality targets, a series of measures have been implemented in the industrial sector, including phasing-out outdated capacity, strengthening emissions standards, and eliminating small, high-polluting factories (Zheng et al., 2018). However, the industry sector still accounted for 35% of the anthropogenic PM2.5-

related premature mortality in 2015. Further reductions by the industrial sector are essential to achieve the air quality targets in the Thirteenth FYP. The residential sector was once the largest contributor to PM2.5-related premature mortality, but its contribution has decreased over time. This decreasing trend was accelerated when millions of households switched from direct coal use to electricity and gas-fired heating during 2013–2017. Wider promotion of clean fuel use in suburban and rural households promises to further reduce emissions from the residential sector.

Although emissions from the power, industrial and residential sectors are decreasing, contributions from the transportation and agricultural sectors have gradually increased. Emissions from the transportation sector have stabilized in recent years because strengthened policy measures have counterbalanced the growing vehicle population. If there were no emission standards during 2000–2012, the vehicle emission levels in China would have increased by 2–6 times the levels measured in 2000 (Huo et al.,



2015). Given that vehicle ownership per thousand people in China is still much lower than that in developed countries, implementing more stringent standards and accelerating the phase-out of older high-emissions vehicles are important to further reduce vehicle emissions. In fact, China is planning to implement the latest National VI emission standards for light-duty vehicles in the air pollution key regions since 2019, and for diesel fueled heavy-duty vehicles since 2021, which foresees even

deeper emission reductions in the near future. Currently, the agricultural sector lacks mitigation policies and foresees growing $NH_3$ emissions in the future. The $NH_3$ emissions growth will offset the air quality benefits from $SO_2$ and $NO_x$ reductions (Wang et al., 2013) and pose challenges for future air quality management. Technologies to improve manure management and reduce ammonia applications in fields and optimizing human diets are potential strategies to mitigate ammonia emissions from the agricultural sector (Zhao et al., 2017). Thus, the emphasis of future mitigations should also be put on controlling vehicle and

agricultural emissions.

## 6 Conclusions

Ambient $PM_{2.5}$ pollution contributed substantially to premature mortality in China. In this study, we investigated the decadal changes of the anthropogenic source contribution of $PM_{2.5}$ air pollution and related health impacts in China. Accompanied with the development of economy and the implementation of environmental policies, the emissions of major air pollutants

changed dramatically during the past 25 years. The $SO_2$, $NO_x$, $PM_{2.5}$, and $NH_3$ emissions increased from 13.6 Mt, 6.4 Mt, 8.9 Mt, and 7.2 Mt in 1990 to 16.9 Mt, 23.7 Mt, 9.1Mt, and 10.5 Mt in 2015, with peaks in 2005, 2010, 2005, and 2015, respectively. In response, the population-weighted $PM_{2.5}$ concentration increased from 36.0 μg m$^{-3}$ in 1990 to 63.5 μg m$^{-3}$ in 2005, then gradually decreased to 49.9 μg m$^{-3}$ in 2015. We estimated that the total premature mortality attributable to anthropogenic $PM_{2.5}$ with GEMM NCD+LRI functions rose from 1.26 million (95% CI: 1.05, 1.46) in 1990 to 2.18 million (95% CI: 1.84, 2.50) in

2005, and fell to 2.10 million (95% CI: 1.76, 2.42) in 2015. Besides the influence from the changes of air quality, the reductions in baseline mortality rates contributed to the decrease of $PM_{2.5}$-related mortality, while the population growth and aging to the increases of $PM_{2.5}$-related mortality.

Investigating the decadal transition in source contribution of $PM_{2.5}$ air pollution and related premature mortality can help to evaluate the effectiveness of past efforts and provide important insights for prioritized strategies in the future. We found that

the contributions of various sectors to anthropogenic $PM_{2.5}$-related premature mortality have changed substantially during 1990–2015. In 1990, the residential sector was the leading source of the $PM_{2.5}$-related mortality (44% of total) in China, followed by industry (29%), power (13%), agriculture (9%) and transportation (5%). Whereas in 2015, the industrial sector became the largest contributor (35%), followed by residential (25%), agriculture (23%), transportation (10%) and power (6%). Limited reducing potential remains in the power sector after effective controls since 2005. The industrial and residential sectors

are still the leading contributors to $PM_{2.5}$-related premature mortality, despite their declining trends. The importance of transportation and agricultural sectors are also highlighted with their continuously increasing contributions. Emphasis should be directed onto the later four sectors when planning future mitigations.





**Data availability**

Data generated from this study are available from the corresponding author upon reasonable request.

**Author contributions**

Q.Z. conceived the study; C.H., M.L., X.L., F.L, D.T., and B.Z. calculated emissions; J.L., Y.Z. and R.W. conducted WRF-
CMAQ simulations; G.G. conducted GEOS-Chem simulations; J.L. conducted estimates of health impacts; Q.Z., J.L, Y.Z.
and D.T. interpreted the data; J.L. and Q.Z. prepared the manuscript with contributions from all co-authors.

**Competing interests**

The authors declare that they have no conflict of interest.

**Acknowledgements**

This work was supported by the National Key R&D program (2016YFC0201506), the National Natural Science Foundation
of China (91744310 and 41625020), the Crystal Tang Foundation, Beijing Natural Science Foundation (8192024), and China
Postdoctoral Science Foundation (2018M641382). We acknowledge Dr. Aaron J. Cohen from Health Effects Institute for
sharing the IER 2015 parameters. We also acknowledge Dr. Randall V. Martin and Dr. Aaron van Donkelaar from Dalhousie
University, and Dr. Yang Liu from Emory University for sharing the satellite-based $PM_{2.5}$ estimates.

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



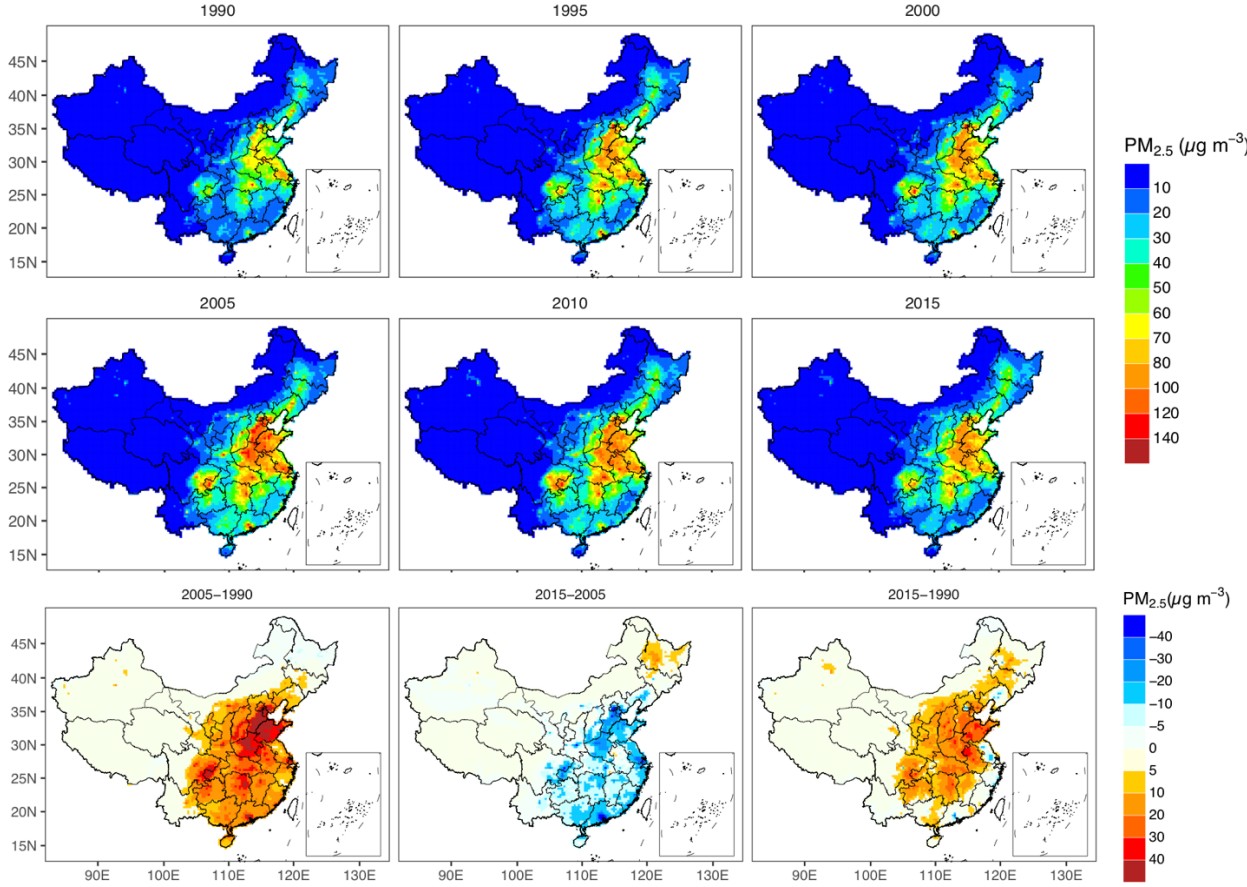

**Figure 1: Annual average PM$_{2.5}$ concentrations and concentration changes from 1990–2015 (µg m$^{-3}$).**



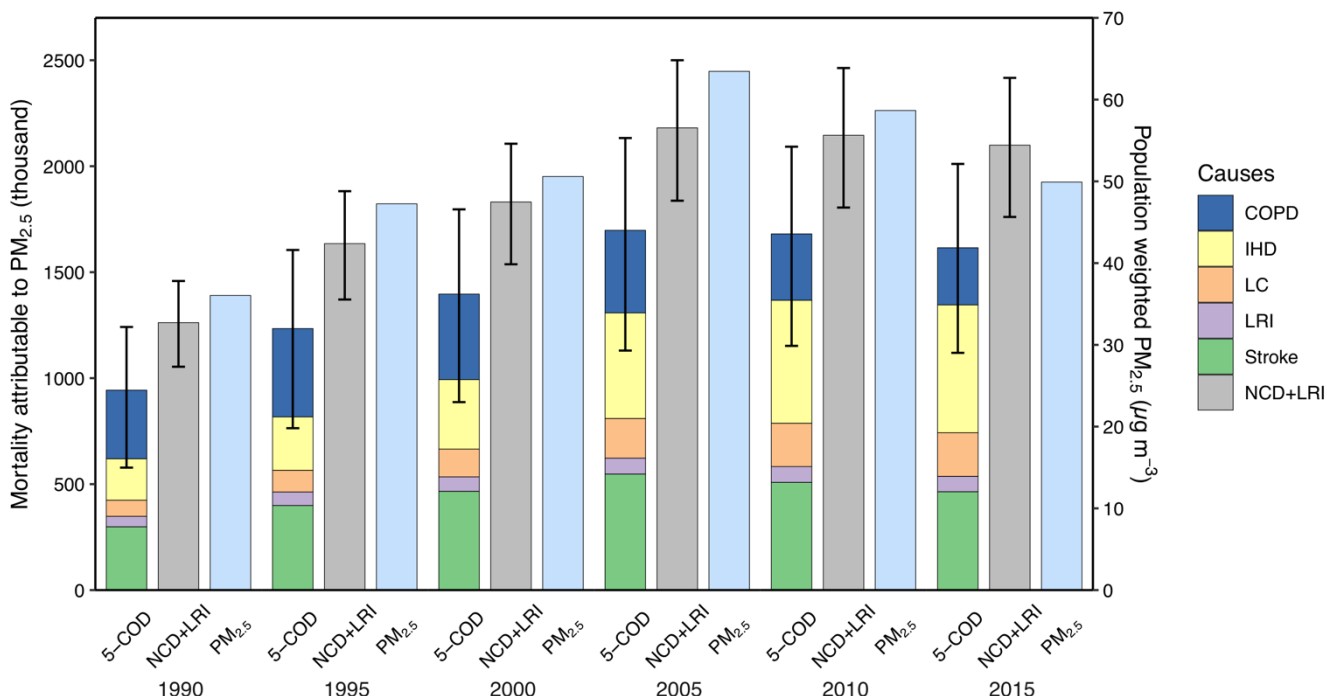

**Figure 2: Historical trends of national population-weighted PM$_{2.5}$ concentrations (µg m$^{-3}$) and PM$_{2.5}$-related premature mortality shared by disease causes (thousand) estimated with GEMM NCD+LRI and GEMM 5-COD functions.**



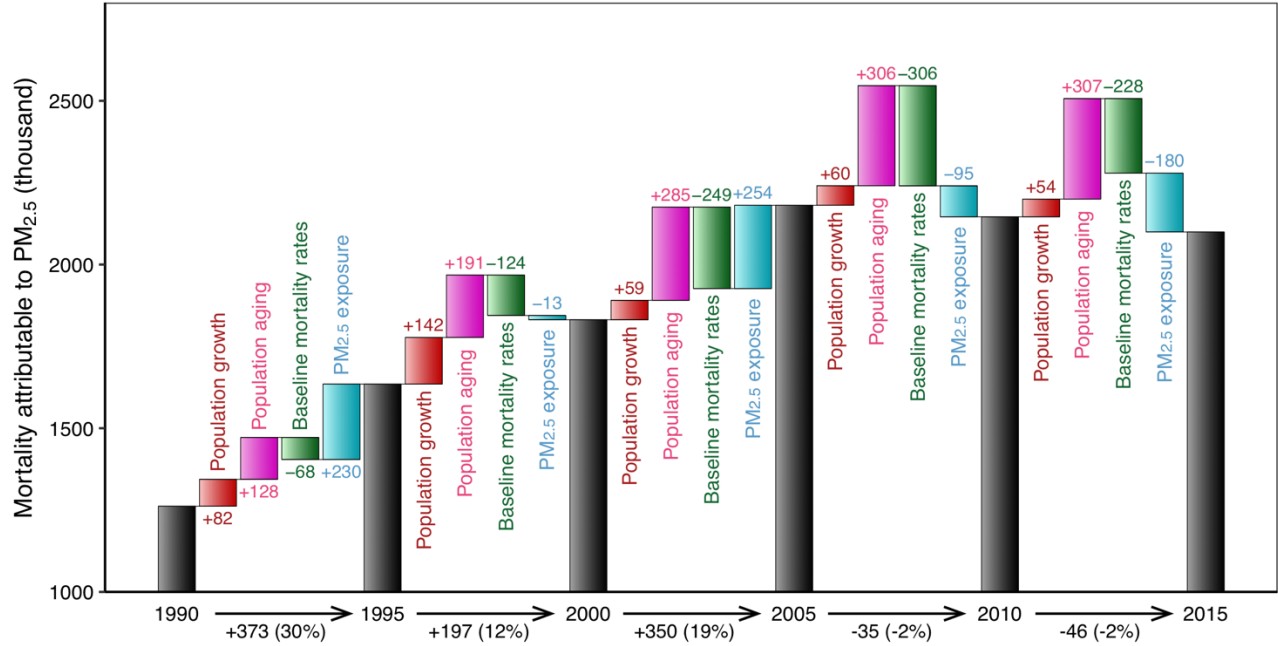

**Figure 3: Contribution of factors to the changes in national PM$_{2.5}$-related premature mortality estimated with GEMM NCD+LRI functions for each five-year interval (in thousands). The length of each bar reflects the contribution of each factor.**







Figure 4: Source contributions to annual average PM$_{2.5}$ concentration (µg m$^{-3}$) in 1990, 2000, 2010 and 2015.





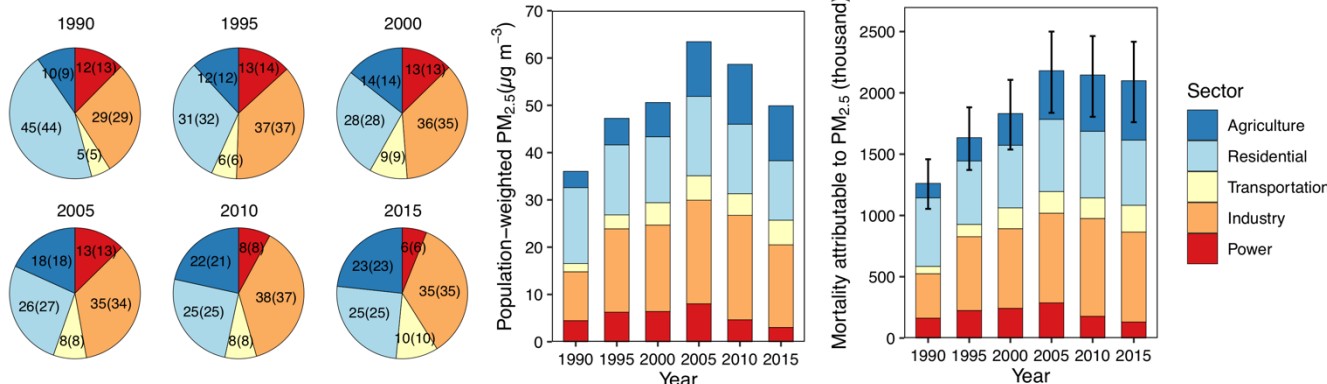

**Figure 5: Relative and absolute source contributions to national population-weighted PM$_{2.5}$ concentrations and related premature mortality estimated with GEMM NCD+LRI functions. a. Relative source contributions (%) to national population-weighted PM$_{2.5}$ concentrations (numbers outside brackets) and related premature mortality (numbers inside brackets). b. Absolute source contributions to national population-weighted PM$_{2.5}$ concentrations (µg m$^{-3}$). c. Absolute source contributions to national PM$_{2.5}$-related premature mortality (thousand) estimated with GEMM NCD+LRI functions.**





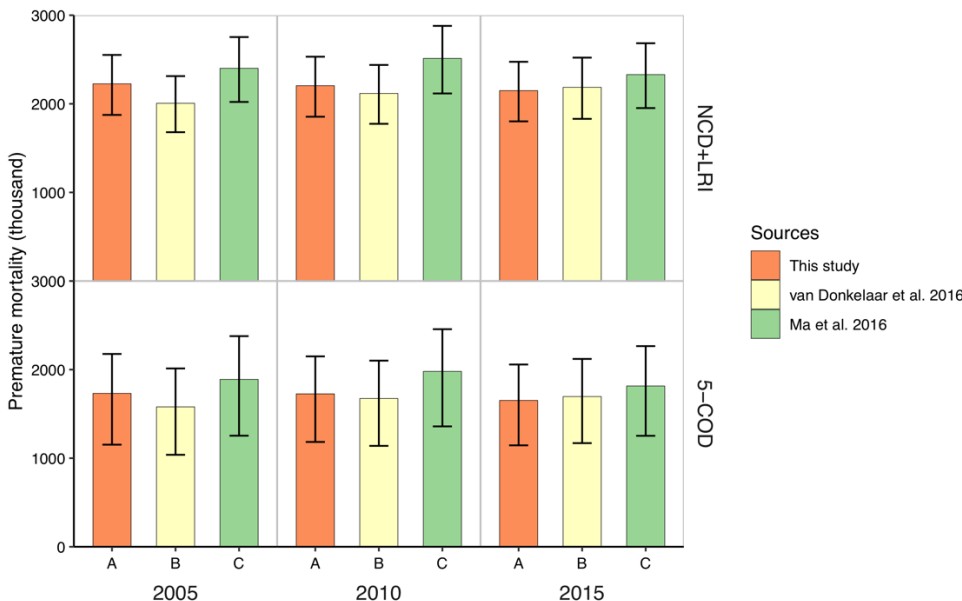

**Figure 6: National PM$_{2.5}$-related premature mortality estimated by PM$_{2.5}$ estimates from CMAQ simulations (this study) and satellite-based methods (thousand).**





**Table 1: Milestones of major air quality regulations in China over the past decades**

| Year | Policy | Target/Measures |
|---|---|---|
| 1987 | Issued *Atmospheric Pollution Prevention Law* | ▪ Aimed at emission control from industry and coal burning |
| 1995 | Revised *Atmospheric Pollution Prevention Law* | ▪ Included acid rain and $SO_2$ pollution controls |
| 2000 | Revised *Atmospheric Pollution Prevention Law* | ▪ Establishing Two Control Zones: Acid Rain Control and $SO_2$ Pollution Control;<br>▪ Implementing $SO_2$ total emission control, vehicle emissions and road dust control in the Two Control Zones. |
| 2005 | Eleventh Five-year Plan | ▪ Reducing national $SO_2$ emissions by 10% |
| 2010 | Twelfth Five-year Plan | ▪ Reducing national $SO_2$ and $NO_x$ emissions by 8% and 10%; |
| 2012 | Ambient Air Quality Standards | ▪ Including $PM_{2.5}$ as an indicator in the standard<br>▪ Holding the annual average concentration of $SO_2$, $NO_2$, CO, $O_3$, $PM_{2.5}$ and $PM_{10}$ below 40 μg m$^{-3}$, 40 μg m$^{-3}$, 4 mg m$^{-3}$, 160 μg m$^{-3}$, 35 μg m$^{-3}$, and 70 μg m$^{-3}$, respectively, to meet the Class Ⅱ standard |
| 2013 | Air Pollution Prevention Action Plan | ▪ By 2017, Reducing the inhalable particle concentration in cities at the prefecture level and above by over 10%;<br>▪ Reducing the $PM_{2.5}$ concentrations in the BTH, YRD and PRD regions by 25%, 15% and 10%, respectively;<br>▪ Controlling the annual $PM_{2.5}$ in Beijing to below 60 μg m$^{-3}$. |
| 2015 | Thirteenth Five-year Plan | ▪ Reducing $PM_{2.5}$ concentration by 18% at the prefecture level and above in cities that failed to meet the $PM_{2.5}$ air quality standard.<br>▪ Reducing national $NH_3$, $SO_2$ and $NO_x$ emissions by 10%, 15% and 15%, respectively;<br>▪ Control VOC emissions in key regions to reduce national emissions by 10%. |



**Table 2: SO$_2$, NO$_x$, PM$_{2.5}$, NH$_3$ emissions, population-weighted PM$_{2.5}$ concentration, and PM$_{2.5}$-related premature mortality contributed by the agricultural (Agr), industrial (Ind), power (Pow), residential (Res) and transportation (Tra) sectors.**

| | | Sector | 1990 | 1995 | 2000 | 2005 | 2010 | 2015 |
|---|---|---|---|---|---|---|---|---|
| | | Agr | 0 (0) | 0 (0) | 0 (0) | 0 (0) | 0 (0) | 0 (0) |
| | | Ind | 5.6 (41) | 8.9 (45) | 8.9 (43) | 13.3 (40) | 16.4 (59) | 9.8 (58) |
| | SO$_2$ | Pow | 5.2 (38) | 7.9 (40) | 9.5 (46) | 16.7 (51) | 7.8 (28) | 3.9 (23) |
| | | Res | 2.8 (21) | 3.0 (15) | 2.3 (11) | 2.8 (9) | 3.4 (12) | 2.9 (17) |
| | | Tra | 0 (0) | 0.1 (0) | 0.1 (1) | 0.2 (1) | 0.2 (1) | 0.3 (2) |
| | | Agr | 0 (0) | 0 (0) | 0 (0) | 0 (0) | 0 (0) | 0 (0) |
| | | Ind | 1.9 (29) | 2.6 (28) | 2.9 (25) | 5.4 (28) | 9.1 (35) | 9.7 (41) |
| | NO$_x$ | Pow | 2 (31) | 3.1 (33) | 3.5 (30) | 6.7 (34) | 8.6 (33) | 5.1 (21) |
| | | Res | 0.8 (12) | 0.8 (8) | 0.7 (6) | 1.0 (5) | 1.0 (4) | 0.9 (4) |
| Emissions [Mt (%)] | | Tra | 1.7 (27) | 3.0 (31) | 4.6 (39) | 6.5 (33) | 7.7 (29) | 8.0 (34) |
| | | Agr | 0 (0) | 0 (0) | 0 (0) | 0 (0) | 0 (0) | 0 (0) |
| | | Ind | 3.4 (39) | 6.3 (52) | 5.7 (51) | 6.8 (50) | 6.1 (52) | 4.4 (48) |
| | PM$_{2.5}$ | Pow | 1.1 (12) | 1.4 (12) | 1.1 (10) | 1.4 (10) | 0.8 (7) | 0.6 (7) |
| | | Res | 4.2 (47) | 4.1 (34) | 3.8 (34) | 4.7 (35) | 4.3 (37) | 3.6 (40) |
| | | Tra | 0.2 (2) | 0.3 (2) | 0.5 (5) | 0.7 (5) | 0.5 (5) | 0.5 (5) |
| | | Agr | 6.7 (93) | 8.3 (94) | 9.0 (94) | 9.3 (93) | 9.5 (93) | 9.7 (93) |
| | | Ind | 0.1 (1) | 0.1 (1) | 0.2 (2) | 0.3 (3) | 0.3 (3) | 0.4 (4) |
| | NH$_3$ | Pow | 0 (0) | 0 (0) | 0 (0) | 0 (0) | 0 (0) | 0 (0) |
| | | Res | 0.4 (5) | 0.4 (4) | 0.3 (3) | 0.4 (4) | 0.4 (4) | 0.3 (3) |
| | | Tra | 0 (0) | 0 (0) | 0 (0) | 0 (0) | 0 (0) | 0 (0) |
| | | Agr | 3.4 (10) | 5.6 (12) | 7.3 (14) | 11.6 (18) | 12.6 (22) | 11.6 (23) |
| Population-weighted PM$_{2.5}$ [µg m$^{-3}$ (%)] | | Ind | 10.3 (29) | 17.6 (37) | 18.3 (36) | 21.9 (35) | 22.1 (38) | 17.5 (35) |
| | | Pow | 4.5 (12) | 6.3 (13) | 6.4 (13) | 8.0 (13) | 4.6 (8) | 3.0 (6) |
| | | Res | 16.1 (45) | 14.8 (31) | 13.9 (28) | 16.8 (26) | 14.7 (25) | 12.6 (25) |
| | | Tra | 1.7 (5) | 2.9 (6) | 4.7 (9) | 5.2 (8) | 4.6 (8) | 5.2 (10) |
| | | Agr | 117.8 (9) [98.5, 136] | 189.9 (12) [159.5, 218.4] | 258.2 (14) [216.9, 296.6] | 396.8 (18) [334.3, 454.7] | 459.0 (21) [386.3, 526.7] | 484.4 (23) [406.5, 557.4] |
| | | Ind | 362.4 (29) [302.6, 418.9] | 602.2 (37) [505.3, 693] | 649.6 (35) [545.7, 746.6] | 731.4 (34) [616.8, 837.5] | 797.7 (37) [671.3, 915.3] | 734.0 (35) [615.5, 844.9] |
| PM$_{2.5}$-related premature mortality {thousand (%) [95%CI]} | NCD+LRI | Pow | 162.3 (13) [135.4, 187.9] | 224.5 (14) [188.0, 258.8] | 242.3 (13) [203.1, 279.2] | 287.0 (13) [241.4, 329.5] | 177.6 (8) [149.1, 204.2] | 130.9 (6) [109.6, 150.9] |
| | | Res | 559.0 (44) [467.0, 645.9] | 517.5 (32) [433.8, 596] | 511.4 (28) [429.1, 588.5] | 589.0 (27) [495.7, 675.8] | 543.4 (25) [456.8, 624.1] | 531.9 (25) [445.9, 612.5] |
| | | Tra | 60.3 (5) [50.4, 69.7] | 100.7 (6) [84.5, 115.9] | 169.9 (9) [142.7, 195.3] | 176.8 (8) [148.9, 202.6] | 168.2 (8) [141.5, 193.1] | 218.3 (10) [183.1, 251.2] |
| | 5-COD | Agr | 88.8 (9) [54.5, 116.6] | 144.2 (12) [89.4, 186.9] | 197.8 (14) [125.7, 253.8] | 309.3 (18) [206, 388.4] | 359.9 (21) [246.9, 447.6] | 373.2 (23) [258.8, 464.2] |



| | | | | | | |
|-----|---|---|---|---|---|---|
| Ind | 270.9 (29) [165.9, 356.6] | 455.0 (37) [281.9, 591.0] | 495.7 (35) [315.2, 636.6] | 569.2 (34) [379.7, 713.3] | 624.9 (37) [428.8, 777.2] | 564.8 (35) [391.4, 703.1] |
| Pow | 120.6 (13) [73.7, 159.4] | 168.9 (14) [104.3, 220.6] | 184.4 (13) [116.7, 238.3] | 223.5 (13) [148.4, 282] | 139.0 (8) [95.1, 173.8] | 100.5 (6) [69.6, 125.4] |
| Res | 417.8 (44) [256.1, 549.5] | 389.8 (32) [241.2, 507.4] | 389.1 (28) [246.9, 501.2] | 457.9 (27) [304.5, 576.6] | 424.9 (25) [291.2, 529.4] | 408.6 (25) [283.2, 508.9] |
| Tra | 45.2 (5) [27.7, 59.4] | 76.1 (6) [47.2, 98.8] | 129.6 (9) [82.4, 166.5] | 137.6 (8) [91.6, 172.7] | 131.8 (8) [90.4, 164.1] | 168 (10) [116.5, 209.1] |





**Table 3: Comparison of anthropogenic source contributions to PM<sub>2.5</sub>-related premature mortality with previous studies (%). The source sectors include Power (Pow), Industry (Ind), Residential (Res), Transportation (Tra), and Agriculture (Agr).**

| Year | Pow | Ind | Res | Tra | Agr | Model | Emission | References |
|------|-----|-----|-----|-----|-----|-------|----------|------------|
| 2010 | 20 | 9 | 36 | 3 | 32 | Global EMAC | EDGAR | Lelieveld et al. 2015 [a] |
| 2010 | 18 | 40 | 22 | 7 | 13 | WRF-CMAQ | HTAP v2 | Gu et al. 2018 [a] |
| 2013 | 13 | 38 | 27 | 7 | 15 | Source-oriented CMAQ | MEIC | Hu et al. 2017 [a] |
| 2014 | 6 | 48 | 42 | 4 | 0.1 | WRF-Chem | EDGAR-HTAP v2 | Reddington. et al. 2019 [a] |
| 2005 | 14 | 33 | 19 | 8 | 26 | WRF-CMAQ | Self-compiled emission | Zheng et al. 2019 [a] |
| 2015 | 7 | 28 | 15 | 15 | 35 | WRF-CMAQ | Self-compiled emission | Zheng et al. 2019 [a] |
| 2005 | 13 | 34 | 27 | 8 | 18 | WRF-CMAQ | MEIC | This study |
| 2010 | 8 | 37 | 25 | 8 | 21 | WRF-CMAQ | MEIC | This study |
| 2015 | 6 | 35 | 25 | 10 | 23 | WRF-CMAQ | MEIC | This study |

a. Relative anthropogenic source contributions are derived from a normalization across all anthropogenic source sectors





**Table 4: PM$_{2.5}$-related premature mortality contributed by the agricultural (Agr), industrial (Ind), power (Pow), residential (Res) and transportation (Tra) sectors. estimated with IER risk function [unit: thousand (%)]**

| Model | Sector | 1990 | 1995 | 2000 | 2005 | 2010 | 2015 |
|-------|--------|------|------|------|------|------|------|
|       | Agr    | 66.6 (9)    | 99.6 (12)   | 127.8 (14)  | 186.1 (18)  | 209.7 (21)  | 217.1 (23)  |
|       | Ind    | 206.7 (29)  | 318.1 (37)  | 323.2 (35)  | 341.9 (33)  | 365.2 (37)  | 330.8 (35)  |
| IER   | Pow    | 93.6 (13)   | 120.2 (14)  | 122.8 (13)  | 136.7 (13)  | 82.8 (8)    | 59.7 (6)    |
|       | Res    | 317.6 (44)  | 274.7 (32)  | 256.5 (28)  | 279.3 (27)  | 250.9 (25)  | 240.4 (25)  |
|       | Tra    | 34.3 (5)    | 53.1 (6)    | 84.5 (9)    | 83.0 (8)    | 77.2 (8)    | 98.2 (10)   |