# Peer review of "Decadal changes in anthropogenic source contribution of PM2.5 pollution and related health impacts in China, 1990–2015"

_Atmospheric Chemistry and Physics, 2019_

## Referee Comment (RC1) · Anonymous Referee #1 · 20 Jan 2020

Driven by the rapid socioeconomic development and environmental policies, air quality in China has changed dramatically over the past few decades. There is a need for studies with long temporal coverage to illustrate the decadal transition of different source sectors. This study investigated the decadal variations of PM2.5 concentrations and the associated health impact, as well as the transition of sectoral contributions, which is quite useful for future control strategies. The topic of this study is therefore valuable and the entire manuscript is well written. I only have a few minor comments.

1) The discussion of anthropogenic emissions: The authors mainly discuss the impact of emissions for the sensitivities. The simulations in 2000, 2005 and 2015 were based

on different meteorological conditions. Although the differences of the sensitivities (i.e., with and without industrial emission) were calculated over the same year (i.e., 2005) first and then the comparison was made across the 25-year, the meteorological differences such as in 2000 and 2005 may yield some differences. In another words, if the same year of meteorology was used for all the simulations, the differences of the sensitivities would not be the same as the current results, meaning that the current results actually reflect the impact from meteorology to a certain extent. Therefore, it is better to add some discussions, i.e., in the uncertainty discussions or the sections related to the sensitivity.

2) NCEP-FNL: the simulations of 1990 and 1995 was based on FNL data in 2000. Actually, besides NCEP-FNL, NCEP-CFSR, ERA-Interim can also be used to drive WRF, and these data was available from 1979. I am not asking the authors to redo the simulations using the reanalysis. In fact, I think it might be useful to add some discussions. For instance, the differences of 1990, 1995 and 2000 really reflect the effect of anthropogenic emissions since the meteorology maintains the same, whereas 2005, 2010 and 2015 reflect the compounded effect from the meteorology on top of the emission. Not sure if this can be quantified by the comparison among two sets of three year simulations (1990, 1995 2000 vs. 2005, 2010, 2015). If this is not possible based on current simulations, some information could be added in the uncertainty discussions.

3) I think it is useful to add some discussions between the policy and the control reflected from the emission inventory. Table 1 listed some policy related to emission control. Table 2 listed the emission inventory, which in a way reflected the control outcome. It can be easily identified that the control such as for SO2 is higher than the planned value (i.e., planned 10% reduction from 2005 to 2010, the SO2 emission in 2005 and 2010 is 33 and 27.8, respectively, with a reduction of 16%). This means that the emission reduction is larger than the planned action. I think it is better to discuss a little bit of this kind of information, not only for SO2, perhaps a few other species as well.

4)Supplement Line 28: S3). In general, the CMAQ simulated anthropogenic PM2.5 has a good correlation with satellite-derived dust-free PM2.5 The authors stated anthropogenic PM2.5. I think the PM2.5 is based on emission from anthropogenic and biogenic emission together. Have the authors done a test by turning off the biogenic emission? Please verify the statement.

———————————————

---

## Referee Comment (RC2) · Anonymous Referee #3 · 28 Feb 2020

China's policy on air pollution control has changed significantly in the past decades, which is expected to influence the air quality and its associated health impact. Certainly, it is of great importance to quantitatively understand the achievement on pollution mitigation and the related health effect. This work is intended to investigate the decadal change of anthropogenic emission sources and the contribution to PM2.5 mitigation since 1990. Based on a bottom-up emission inventory, regional chemical transport model, and the exposure mortality estimation, this study quantified the change in anthropogenic emission intensity and the resultant mortality due to PM2.5 variations. Overall, this manuscript is well structured and also well written. It also provided some possible policy implementation for pollution control. Here are some comments to be

addressed, after which I think it is worth publishing in this journal.

Apart from PM2.5, ozone also has great negative effects on human health, worsening chronic respiratory diseases such as asthma and compromise the ability of the body to fight respiratory infections. As emissions change in the past decades, ozone level varies in response, and so does its influence on health. Have the premature mortality data or model used in this study considered it?

In section 3, it is a little bit tedious to just describe China's air pollution regulations. Also, more analysis on pollution regulation change and long-term emission variation may improve the clarity. Thus a timeline chart of both pollution regulation and emission inventory is suggested to be added here.

Page 1 Line 17: Does "CI" stand for "Confidence Interval" here? Full expression is recommended for the first time of the statement.

Page 2 Line 12: Quantification of the pollution level would be more rigorous, like how many times of average PM concentration in the area compared with WHO or China national standard?

---

## Author Comment (AC1) · 8 May 2020

Driven by the rapid socioeconomic development and environmental policies, air quality in China has changed dramatically over the past few decades. There is a need for studies with long temporal coverage to illustrate the decadal transition of different source sectors. This study investigated the decadal variations of $PM_{2.5}$ concentrations and the associated health impact, as well as the transition of sectoral contributions, which is quite useful for future control strategies. The topic of this study is therefore valuable and the entire manuscript is well written. I only have a few minor comments.

**Response:** We thank the Referee for the insightful comments and positive tones. We have revised the manuscript according to the suggestions and respond to the concerns below.

1. The discussion of anthropogenic emissions: The authors mainly discuss the impact of emissions for the sensitivities. The simulations in 2000, 2005 and 2015 were based on different meteorological conditions. Although the differences of the sensitivities (i.e., with and without industrial emission) were calculated over the same year (i.e., 2005) first and then the comparison was made across the 25-year, the meteorological differences such as in 2000 and 2005 may yield some differences. In another words, if the same year of meteorology was used for all the simulations, the differences of the sensitivities would not be the same as the current results, meaning that the current results actually reflect the impact from meteorology to a certain extent. Therefore, it is better to add some discussions, i.e., in the uncertainty discussions or the sections related to the sensitivity.

**Response:** Accepted. We've added three sensitivity simulations to estimate the influence from meteorology, and added discussions below as part of Uncertainties and Limitations in Section 5.2:

*"To quantify the influence from the interannual variations of meteorology, we carried out three sensitivity simulations with the same meteorology in year 2000 and the year-specific emissions in 2005, 2010 and 2015. The differences between the base and the*

*sensitivity simulation denote the impacts of meteorology. It shows that the impacts of the interannual variations of meteorological condition differ with regions and leads to an overall change of the population-weighted PM$_{2.5}$ concentration below 5%. The interannual changes of PM$_{2.5}$ concentrations were dominated by the changes of emissions."*

2. NCEP-FNL: the simulations of 1990 and 1995 was based on FNL data in 2000. Actually, besides NCEP-FNL, NCEP-CFSR, ERA-Interim can also be used to drive WRF, and these data was available from 1979. I am not asking the authors to redo the simulations using the reanalysis. In fact, I think it might be useful to add some discussions. For instance, the differences of 1990, 1995 and 2000 really reflect the effect of anthropogenic emissions since the meteorology maintains the same, whereas 2005, 2010 and 2015 reflect the compounded effect from the meteorology on top of the emission. Not sure if this can be quantified by the comparison among two sets of three-year simulations (1990, 1995 2000 vs. 2005, 2010, 2015). If this is not possible based on current simulations, some information could be added in the uncertainty discussions.

**Response:** Accepted. Initially, we tried to combine the NCEP-CFSR for 1990-1995 and NCEP-FNL for 2000-2015 to drive the WRF model, since the NCEP-CFSR was only available for the period of 1979 to 2010, and the NCEP-FNL was only available since 1999. However, we found that there were non-negligible differences in the simulations driven by the two products. We ran the WRF-CMAQ simulation for the overlapping year 2000 driven by both NCEP-CFSR and NCEP-FNL, and found that the differences in the chemical composition of the population-weighted PM$_{2.5}$ could be as high as 1.0 μg m$^{-3}$, which may introduce extra bias to the trend of relative source contributions. Therefore, we used the same product to drive the WRF model. We have added the reviewer's concern as a part of uncertainty in Section 5.2 and added three sensitivity simulations to quantity the influence from variations in meteorology:

*"Third, uncertainties are also introduced by the meteorological conditions. In the study, we applied the NCEP-FNL meteorological data in 2000 to drive the WRF model for year 1990 and 1995. Initially, we intended to combine the NCEP Climate Forecast*

*System Reanalysis (NCEP-CFSR) for 1990-1995 and NCEP-FNL for 2000-2015, since the NCEP-CFSR was available before 2011, and the NCEP-FNL was available after 1999. However, when we ran the WRF-CMAQ model for the overlapping year 2000 with both products, we found the differences in the chemical composition of the population-weighted $PM_{2.5}$ could be as high as 1.0 µg $m^{-3}$, which may introduce extra bias to the trend of relative source contributions. Therefore, we applied the same meteorological product to drive the WRF model. To quantify the influence from the interannual variations of meteorology, we added three sensitivity simulations with the same meteorology in year 2000 and the year-specific emissions in 2005, 2010 and 2015. The differences between the base and the sensitivity simulation denotes the impacts of meteorology. It shows that the impacts of the interannual variations of meteorological condition differ with regions and leads to an overall change of the population-weighted $PM_{2.5}$ concentration below 5%. The interannual changes of $PM_{2.5}$ concentrations were dominated by the changes of emissions."*

3. I think it is useful to add some discussions between the policy and the control reflected from the emission inventory. Table 1 listed some policy related to emission control. Table 2 listed the emission inventory, which in a way reflected the control outcome. It can be easily identified that the control such as for $SO_2$ is higher than the planned value (i.e., planned 10% reduction from 2005 to 2010, the $SO_2$ emission in 2005 and 2010 is 33 and 27.8, respectively, with a reduction of 16%). This means that the emission reduction is larger than the planned action. I think it is better to discuss a little bit of this kind of information, not only for $SO_2$, perhaps a few other species as well.

**Response:** Accepted. Thanks to the referee's suggestion, it's better to integrate the policies and regulations with the effects in emission changes. We have rearranged the paragraph with additional information on the changes of emissions in response to policies and regulations in Section 3:

*"Table 1 lists the development sequence of the major air quality regulations in China, Figure S4 shows the timetable of the emissions standards implemented in the major*

*sectors during past decades, and Table 2 presents the annual emissions of $SO_2$, $NO_x$, $PM_{2.5}$, and $NH_3$ contributed by agriculture, industry, power, residential and transportation sectors……However, these measures did not keep up with the rapid growth of economy and fossil fuel use, and the national emissions $SO_2$, $NO_x$ and $PM_{2.5}$ had increased by 142%, 207%, and 54% from 1990 to 2005……. In response, the national $SO_2$ emission dropped from 33.0 Mt to 27.8Mt, by 16% in period of 2005-2010, which was even greater than the target reduction rate of 10%. But the $NO_x$ emission kept growing during the Eleventh FYP due to limited end-of-pipe measures, and started to drop during the Twelfth FYP……As a result, the $SO_2$, $NO_x$ and $PM_{2.5}$ emissions in 2015 had decreased dramatically by 39%, 10% and 22%, compared with the levels in 2010."*

4. Supplement Line 28: S3). In general, the CMAQ simulated anthropogenic $PM_{2.5}$ has a good correlation with satellite-derived dust-free $PM_{2.5}$ The authors stated anthropogenic $PM_{2.5}$. I think the $PM_{2.5}$ is based on emission from anthropogenic and biogenic emission together. Have the authors done a test by turning off the biogenic emission? Please verify the statement.

**Response:** Yes, we have excluded the contribution of biogenic emission by one additional "clean" simulation. As stated in the last paragraph of Section 2.2, "Since we focused on the contribution and relative importance of anthropogenic source sectors, we conducted another "clean" simulation for each year by removing all the Chinese anthropogenic emissions in the total model emissions, to exclude the contributions of boundary conditions, and emissions from biogenic sources, dust, sea salt, and other countries."

---

## Author Comment (AC2) · 8 May 2020

China's policy on air pollution control has changed significantly in the past decades, which is expected to influence the air quality and its associated health impact. Certainly, it is of great importance to quantitatively understand the achievement on pollution mitigation and the related health effect. This work is intended to investigate the decadal change of anthropogenic emission sources and the contribution to PM2.5 mitigation since 1990. Based on a bottom-up emission inventory, regional chemical trans- port model, and the exposure mortality estimation, this study quantified the change in anthropogenic emission intensity and the resultant mortality due to PM2.5 variations. Overall, this manuscript is well structured and also well written. It also provided some possible policy implementation for pollution control. Here are some comments to be addressed, after which I think it is worth publishing in this journal.

**Response:** We appreciate the Referee's positive tone. Below we have point-by-point addressed the Referee's comments.

1. Apart from $PM_{2.5}$, ozone also has great negative effects on human health, worsening chronic respiratory diseases such as asthma and compromise the ability of the body to fight respiratory infections. As emissions change in the past decades, ozone level varies in response, and so does its influence on health. Have the premature mortality data or model used in this study considered it?

**Response:** Besides $PM_{2.5}$, ozone also has adverse health effects. In this study, we restricted our focus on the source contribution of $PM_{2.5}$ pollution and related health, and have not considered the impact of ozone yet. The health effects of ozone can be analyzed in another study in the future. We clarified our research focus in Section 2.3: "*Besides PM$_{2.5}$, ozone also has adverse health effects. In this study, we focus on the premature mortality due to the long-term PM$_{2.5}$ exposure and did not consider the impact of ozone.*"

2. In section 3, it is a little bit tedious to just describe China's air pollution regulations.

Also, more analysis on pollution regulation change and long-term emission variation may improve the clarity. Thus, a timeline chart of both pollution regulation and emission inventory is suggested to be added here.

**Response:** Accepted. Referee #1 also suggested to add discussion on the changes of emission in response to policies. Therefore, we have rearranged the paragraph in Section 3 with additional information on the changes of emissions from Table 2:

*"Table 1 lists the development sequence of the major air quality regulations in China, Figure S4 shows the timetable of the emissions standards implemented in the major sectors during past decades, and Table 2 presents the annual emissions of $SO_2$, $NO_x$, $PM_{2.5}$, and $NH_3$ contributed by agriculture, industry, power, residential and transportation sectors……However, these measures did not keep up with the rapid growth of economy and fossil fuel use, and the national emissions $SO_2$, $NO_x$ and $PM_{2.5}$ had increased by 142%, 207%, and 54% from 1990 to 2005……. In response, the national $SO_2$ emission dropped from 33.0 Mt to 27.8Mt, by 16% in period of 2005-2010, which was even greater than the target reduction rate of 10%. But the $NO_x$ emission kept growing during the Eleventh FYP due to limited end-of-pipe measures, and started to drop during the Twelfth FYP……As a result, the $SO_2$, $NO_x$ and $PM_{2.5}$ emissions in 2015 had decreased dramatically by 39%, 10% and 22%, compared with the levels in 2010."*

3. Page 1 Line 17: Does "CI" stand for "Confidence Interval" here? Full expression is recommended for the first time of the statement.

**Response:** Accepted. "CI" stands for "Confidence Interval". We have revised it as "1.26 million [95% Confidential Interval (95% CI): 1.05, 1.46]" for the first time of the statement.

4. Page 2 Line 12: Quantification of the pollution level would be more rigorous, like how many times of average PM concentration in the area compared with WHO or China national standard?

**Response:** Accepted. We have quantified the pollution level by comparison with China

national standard and added information below for clarify.

*"During 2013-2014, only three out of the thirty-one provincial capital cities in China had the PM$_{2.5}$ annual concentration below the national standard grade II of 35 μg m$^{-3}$, and the highest concentration of 144 μg m$^{-3}$ occurred Shijiazhuang, which were over three times higher than the standard (Wang et al., 2014)."*